# Evolution and Functional Characteristics of the Novel *elovl8* That Play Pivotal Roles in Fatty Acid Biosynthesis

**DOI:** 10.3390/genes12081287

**Published:** 2021-08-23

**Authors:** Shouxiang Sun, Yumei Wang, Pei-Tian Goh, Mónica Lopes-Marques, L. Filipe C. Castro, Óscar Monroig, Meng-Kiat Kuah, Xiaojuan Cao, Alexander Chong Shu-Chien, Jian Gao

**Affiliations:** 1Key Lab of Freshwater Animal Breeding, College of Fisheries, Ministry of Agriculture, Huazhong Agricultural University, Wuhan 430070, China; ssx0707@webmail.hzau.edu.cn (S.S.); YumeiWang@webmail.hzau.edu.cn (Y.W.); caoxiaojuan@mail.hzau.edu.cn (X.C.); 2Ministry of Education/Hubei Provincial Engineering Laboratory for Pond Aquaculture, Engineering Research Center of Green Development for Conventional Aquatic Biological Industry in the Yangtze River Economic Belt, College of Fisheries, Huazhong Agricultural University, Wuhan 430070, China; 3Center for Chemical Biology, Universiti Sains Malaysia, Bayan Lepas 11800, Penang, Malaysia; gohpeitian@gmail.com (P.-T.G.); kuahmk@gmail.com (M.-K.K.); 4CIIMAR/CIMAR—Interdisciplinary Center of Marine and Environmental Research, University of Porto, 4099002 Porto, Portugal; monicaslm@hotmail.com (M.L.-M.); filipe.castro@ciimar.up.pt (L.F.C.C.); 5Department of Biology, Faculty of Sciences of the Univeristy of Porto (FCUP), 4099002 Porto, Portugal; 6Instituto de Acuicultura Torre de la Sal (IATS-CSIC), Ribera de Cabanes, 12595 Castellon, Spain; oscar.monroig@csic.es

**Keywords:** *elovl8*, gene knockout, zebrafish, fatty acid synthesis

## Abstract

Elongation of very long-chain fatty acid (Elovl) proteins are key enzymes that catalyze the rate-limiting step in the fatty acid elongation pathway. The most recently discovered member of the Elovl family, Elovl8, has been proposed to be a fish-specific elongase with two gene paralogs described in teleosts. However, the biological functions of Elovl8 are still to be elucidated. In this study, we showed that in contrast to previous findings, *elovl8* is not unique to teleosts, but displays a rather unique and ample phylogenetic distribution. For functional determination, we generated *elovl8a* (*elovl8a^−^*^/*−*^) and *elovl8b* (*elovl8b^−^*^/*−*^) zebrafish using CRISPR/Cas9 technology. Fatty acid composition in vivo and zebrafish liver cell experiments suggest that the substrate preference of Elovl8 overlapped with other existing Elovl enzymes. Zebrafish Elovl8a could elongate the polyunsaturated fatty acids (PUFAs) C18:2n-6 and C18:3n-3 to C20:2n-6 and C20:3n-3, respectively. Along with PUFA, zebrafish Elovl8b also showed the capacity to elongate C18:0 and C20:1. Gene expression quantification suggests that Elovl8a and Elovl8b may play a potentially important role in fatty acid biosynthesis. Overall, our results provide novel insights into the function of Elovl8a and Elovl8b, representing additional fatty acid elongases not previously described in chordates.

## 1. Introduction

Fatty acids (FAs) including saturated and unsaturated fatty acids have been shown to have significant roles in numerous critical biological processes and are also major components of complex lipid molecules. Upon the formation of palmitic acid (16:0), further modification involves the catalytic activities of fatty acid desaturase (Fads) and fatty acid elongases (Elovls) [1,2]. The latter extend the carbon chain of fatty acid through adding two carbon units at the carboxyl end in the endoplasmic reticulum, which has a rate-limiting effect in the process of fatty acid synthesis. In mammals, seven different families of the Elovls, termed Elovl1–Elovl7, have been identified [3].

The seven members of the Elovl family can be divided into two major groups on the basis of their substrate specificity. While Elovl1, Elovl3, Elovl6, and Elovl7 are implicated in the elongation of saturated fatty acids (SFAs) and monounsaturated fatty acids (MUFAs), members of the Elovl2, Elovl4, and Elovl5 groups are shown to participate in the elongation of polyunsaturated fatty acids (PUFAs) [4,5,6,7]. More specifically, Elovl1 mainly elongates SFA and MUFA C20- and C22-acyl coenzyme A (CoA) [8]. Elovl3 has elongation activity for SFA and MUFA C18- to C22-CoAs [9]. The Elovl6 catalyzes the chain elongation of C16:0 to C18:0 and C16:1 to C18:1, respectively [7,10,11]. Elovl7 has been reported to be involved in the elongation of (C18–C22) SFAs, C18:1 and C18 PUFAs, 18:3n-3 [12,13]. The interest in deciphering the capacity of aquaculture fish species to biosynthesize the physiologically important long-chain polyunsaturated fatty acids (LC-PUFA) has led to widespread efforts to characterize Elovl2 and Elovl5 in a myriad of species [14,15,16,17,18,19,20]. The majority of the teleost Elovl5 possesses a preference toward C18 PUFA and C20 substrates, while Elovl2 is inclined to elongate C22 substrates [14,21]. Several exceptions to this dichotomy have also been reported [22,23]. Additionally, while Elovl5 orthologs are widespread across multiple taxa groups, Elovl2 has only been reported from salmonids and several teleost families [15,16,17,20,24,25]. Elovl4 was shown to play a role in the elongation of very long-chain fatty acids (VLC-FAs) such as >C24 SFAs and PUFAs in vertebrates [4,26,27]. Additionally, teleost Elovl4 could also catalyze the elongation of C22 PUFA substrates to C24 products, leading to the speculation that this ortholog could compensate for the loss of Elovl2 in many marine species [28].

A previous work on the Atlantic cod (*Gadus morhua*) reported the cloning of two putative *elovl* that was categorized as *elovl4c*, despite their separation from the functionally proven Elovl4 clades [29]. A broader search revealed similar orthologs from Atlantic salmon (*Salmo salar*), channel catfish (*Ictalurus punctatus*), and tilapia (*Oreochromis niloticus*), which were all annotated as *elovl4* or *elovl4*-like. Subsequently two zebrafish (*Danio rerio*) isoforms of this ortholog were termed as *elovl8a, b* and deposited in GenBank [30]. More recently, full length cDNA of *elovl8a* and *elovl8b* were cloned from rabbitfish (*Siganus caniculatus*) and characterized for their in vitro elongation capacity on different FAs through functional in vitro assay employing heterologous yeast expression [31]. Results showed *elovl8a* had no activity and *elovl8b* had lower activity in the elongation of C18 and C20 PUFA substrates. However, this approach could be limited by the codon preference of yeast or the inability to adequately replicate the FA elongation process of fish cells. Therefore, the precise role of Elovl8s in vivo regarding LC-PUFA biosynthesis is unclear, and further functional analysis using alternative methods is necessary.

The zebrafish is a reliable model to decipher roles of FA metabolism genes during development due to the presence of numerous conserved homolog genes encoding for the lipid and lipoprotein metabolism pathways [32,33,34]. Specific to LC-PUFA biosynthesis, the first bifunctional desaturase, a Fads2 with Δ5 and Δ6 activities, was reported from zebrafish [35]. Subsequently, the functional activities, dietary regulation, and expression profile of *fads2* and various *elovl* paralogs in different tissue and developmental stages were studied using zebrafish [18,19,36,37,38,39]. In addition, in vivo aspects of the transcriptional regulation *fads2* and *elovl5* were also investigated using zebrafish [40,41]. To understand the specific physiological function of Elovl8, we explored the evolution of Elovl8 in vertebrates and investigated the functional characteristics of zebrafish *elovl8a* and *elovl8b*, and generated CRISPR/Cas9-mediated knockout zebrafish models for the first time.

## 2. Materials and Methods

### 2.1. Ethics

This study was conducted in strict accordance with the recommendations in the guide for the care and use of laboratory animals of Huazhong Agricultural University. This study was approved by the Committee on the Ethics of Animal Experiments of Huazhong Agricultural University (HZAUFI-2021-0022). To minimize suffering, zebrafish were killed after anesthesia with MS-222 (tricaine methanesulfonate, Sigma, St. Louis, MO, USA).

### 2.2. Phylogenetic Analysis and Synteny Maps

Elovl1, Elovl7, Elovl4, and Elovl8 amino acid sequences were sampled from all major vertebrate lineages with genomes available (mammals, reptiles, birds, amphibians, coelacanthiformes, teleosts, and chondrichthyans) and the invertebrate deuterostome and protostome species *Ciona intestinalis, Branchiostoma floridae, Branchiostoma belcheri*, and *Octopus vulgaris* (accession numbers available in Appendix A). Sequences were collected using the ortholog pipeline in Ensembl genome browser or by blastp and/or tblastn in NCBI and Skatebase (http://skatebase.org/, accessed data 24 June 2019) [42]. Collected sequences were inspected and partial sequences were removed, leaving a total of 89 sequences, which were subsequently aligned using MAFFT-L-INS-i (v7.452) available at https://mafft.cbrc.jp/alignment/server/ (accessed data 24 June 2019) [43,44,45]. The resulting sequence alignment was uploaded into Geneious R7.1.9 and all columns containing 90% gaps were stripped from the alignment, leaving a total of 328 positions for phylogenetic analysis. Phylogenetic analyses were performed in PHYML 3.0 [46] server available at http://www.atgc-montpellier.fr/phyml/ (accessed data 24 June 2019). The evolutionary model was determined using the Smart Model Selection (SMS) option [47] resulting in a JTT + G + I + F and branch support was calculated using aBayes posterior probabilities [48]. The final tree was analyzed using FigTreev1.3.1 available at http://tree.bio.ed.ac.uk/software/figtree/ (accessed data 24 June 2019).

Comparative synteny maps were constructed using the latest genome assemblies available in the Gene database at NCBI, namely *Homo sapiens* (GRCh38.p13), *Bos taurus* (ARS-UCD1.2) *Anas platyrhynchos* (IASCAAS_PekingDuck_PBH1.5), *Chrysemys picta* (Chrysemys picta_bellii-3.0.3), *Xenopus tropicalis* (UCB_Xtro_10.0), *Latimeria chalumnae* (LatCha1), *D. rerio* (GRCz11), *Lepisosteus oculatus* (LepOcu1), and *Callorhinchus milii* (Callorhinchus_milii-6.1.3). Maps were created and centered on the target gene *Elovl8* with four protein coding neighboring genes collected to each side, where possible. The phylogenetic relationships of flanking genes were deduced from the Ensembl tree pipeline.

### 2.3. Zebrafish Maintenance and Feed Experiment

Wild-type (WT) zebrafish (AB strain, obtained from the Institute of Hydrobiology, Chinese Academy of Sciences, China Zebrafish Resource Center, Wuhan, China) were raised in 28 °C circulating water with the photoperiod of 14L:10D. Zebrafish were raised to two-months-old and fed with brine shrimp *Artemia* sp. at a fixed ratio. The feeding amounts were correspondently adjusted over time. The average brine shrimp intake by zebrafish was estimated at 6 mg/fish/day.

The experiment diets were prepared using a commercial feed (nutrient compositions: 35% crude protein, 3% crude lipid, 10% ash, and 4% fiber, PIKE Biotechnology Co. Ltd., Beijing, China). Finely ground feeds (50 g) were supplemented with 200 mg of each of the following FAs: (C18:0, C20:0, C18:3n-3) by mixing the feed particles with a solution of FAs in methanol. After air drying, experimental diets were stored at −20 °C until further use. A total of 180 WT zebrafish (15 fish/ tank in triplicate) with uniform-size (0.16 ± 0.01 g) were divided into four groups control (the commercial feed), C18:0 diet, C20:0 diet, and C18:3n-3 diet). The zebrafish from four groups were fed at a fixed ration (20 mg/fish/day). After four weeks of feeding, the fish were starved for 24 h before sampling. Livers of the zebrafish (n = 3 for each group, samples from three fishes mixed into a biological sample) were sampled for qPCR assays.

### 2.4. ZFL Cell Culture and Treatments

The zebrafish liver cell line (ZFL) (obtained from the China Zebrafish Resource Center, Wuhan, China) were maintained in modified limit dilution factor (LDF) medium (50% Leibowitz-15, 35% Dulbecco’s modified Essential medium, 15% HAM’s F12, 15 mM HEPES, 0.15 g/L NaHCO_3_, and 10 μg/mL bovine insulin) supplemented with 5% (*v*/*v*) fetal bovine serum (FBS, Gibco) and 2% antibiotic (100 U/mL penicillin, 100 μg/mL streptomycin), and kept at 28 °C in a 0.5% CO_2_ incubator [49].

The detailed methods of the preparation of FA solutions were performed as previously described with some modifications [50]. Briefly, aliquots (500 μL) of a methyl-β-cyclodextrin (MβCD) solution in water were added to microcentrifuge tubes containing one of the following SFAs (purity greater than 99%): C14:0, C16:0, C18:0, C20:0, and C22:0 followed by incubation at 70 °C for 30 min and sonication for 5 min to obtain the stock solution at 2 mM. A stock solution of MβCD alone was made in water at 100 mM. All stock solutions of MβCD alone were made in water at 100 mM. After incubation at 70 °C for 1 h and sonication for 5 min, all stock solutions of FA:MβCD were clear at room temperature. For PUFAs, the stock solutions of C18:2n-6, C18:3n-3, C18:4n-6, C20:3n-3, C20:4n-6, C22:5n-3, and C22:6n-3 (2 mM) were prepared in ethanol. The ZFL cell were incubated with either SFAs or PUFAs at 50 μM in triplicate per treatment. All FAs used in this study were purchased from Aladdin Co. Ltd., Shanghai, China.

Both *elovl8a* and *elovl8b* knockdown in ZFL cells was performed by siRNA transfection. The siRNA sequences are shown in Appendix A. The siRNA was synthetized by Shanghai GenePharma Co. Ltd. (Suzhou, China). Lipofectamine 3000 reagent was used for cell transfection according to the manufacturer’s instructions (Invitrogen, Carlsbad, CA, USA). The final concentration of siRNA was 50 nM. The transfected cells were cultured at 28 °C for 24 h. Using at least 10^6^ cells, aspirate the media and wash once with ice cold PBS for RNA isolation.

### 2.5. qPCR Analysis

Total RNA was extracted from ZFL cells and liver tissue of zebrafish using the RNA isoPius Kit following the manufacturer’s instructions (TaKaRa, Kyoto, Japan). Quality and quantity of isolated RNA were checked by electrophoresis and spectrophotometry (A260/A230 ratio of around 2 or slightly above; A260/A280 ratio between 1.8–2.0) prior to reverse transcription into complementary DNA (cDNA) with the PrimeScript RT Reagent Kit (TaKaRa, Kyoto, Japan). Working solutions of 1:5 diluted cDNA in ddH_2_O were prepared. HieffTM qPCR SYBR Green Master Mix was purchased from YEASEN Biotech Co. Ltd. (Shanghai, China). Parameters for qPCR runs were as follows: step 1: 95 °C for 5 min (heating rate: 1.6 °C/s); step 2: 40 cycles of 95 °C for 10 s, 55–60 °C for 20 s; 72 °C for 20 s (heating rate: 1.6 °C/s); step 3: 95 °C for 15 s, 60 °C for 60 s and 95 °C for 15 s (heating rate: 0.15 °C/s). The 2^−ΔΔCT^ method was used to analyze the expression levels of the target genes with *β-actin* and glyceraldehyde-3-phosphate dehydrogenase (*gapdh*) used as reference genes. The primer sequences for qPCR are listed in Appendix A. All procedures were based on the methods described by Liang et al. [51].

### 2.6. Fatty Acid Composition Analysis

Total lipids from ZFL cells (4 × 10^6^) and livers (20 mg) of WT, *elovl8a^−^*^/*−*^, and *elovl8b^−^*^/*−*^ zebrafish were extracted using the Bligh and Dyer procedure as previously described [52]. Then, the total lipid was methylated for 1.5 h at 85 °C with 3 mL methylation reagent containing 1% H_2_SO_4_ (*v*/*v*) and 99% methylation and 0.01% (*w*/*v*) butylated hydroxytoluene to produce fatty acid methyl esters (FAMEs). Methyl tricosanoate (Nu-Chek prep. Inc., Elysian, MN, USA) was used as an internal standard at 1.0 mg/mL hexane. FA composition of SFA, MUFA, and PUFA fractions were determined using gas chromatography (Agilent Technologies, Inc, Santa Clara, CA, USA; column: OmegawaxTM320) according to the method of Sun et al. [18]. The temperature of the injector and detector were set at 250 °C and 260 °C, respectively. The temperature program was 200 °C (40 min) to 240 °C (15 min) at 4 °C/min. High purity helium was used as the carrier gas at a flow rate of 1 mL/min. The samples (1.0 μL) were automatically injected into the injection port and identified FAs were presented as area percentage of total FAs.

### 2.7. In Situ Hybridization of Zebrafish Embryos

Spatial expression of the zebrafish *elovl8a* and *elovl8b* in 96 h post-fertilization (hpf) embryos was examined by using the whole mount in situ hybridization (WISH) protocol as described earlier [53]. The cDNA fragments of zebrafish *elovl8a* and *elovl8b* were used as templates for in vitro transcription to synthesize riboprobes for WISH staining. Specific primers were designed to synthesize the *elovl8a* and *elovl8b* riboprobes (Appendix A). Stained embryos were mounted in glycerol and observed under a Research Macro Zoom Fluorescence Microscope (Olympus MVX10 MacroView, Olympus Corp., Shanghai, China) and photographs were taken with an Olympus SCX10 camera.

### 2.8. Elovl8a and Elovl8b Gene Knockout by CRISPR/Cas9 in Zebrafish

In order to investigate the functions of *elovl8* in zebrafish, we generated two knockout models, namely *elovl8a^−^*^/*−*^ and *elovl8b^−^*^/*−*^. The DNA-sequence of *elovl8a* and *elovl8b* were obtained from NCBI (http://www.ncbi.nlm.nih.gov/, accessed data 24 June 2019) and the target gene regions were amplified using the primers for *elovl8a* and *elovl8b* (Appendix A). In vitro transcription of Cas9 RNA and gRNA were based on the standards of the relevant research [54]. Detailed construction methods were performed as previously described [55]. The genomic DNA was isolated from six randomly selected fertilized embryos. Next, the target genome region was amplified and sequenced. Once the mutation was confirmed in injected embryos, the remaining ones were raised to adulthood and the mutant ones were outcrossed with WT zebrafish to produce F1 generation. Two months later, the heterozygous F1 generation with the same mutation sequences was confirmed by sequencing the genomic DNA from the cut tail fin and was self-crossed. About a quarter of the F2 generation obtained were homozygous mutants. The F3 individuals of two-month-old (namely, KO zebrafish) produced by self-crossing F2 homozygous mutants were used in this study. The *elovl5^−^*^/*−*^ [18], *elovl1a^−^*^/*−*^, *elovl1b^−^*^/*−*^, and *elovl3b^−^*^/*−*^ zebrafish were previously generated in our lab.

### 2.9. Statistical Analysis

Statistical analyses were conducted by *T* tests in IBM SPSS statistics 22 software (SPSS Inc., Chicago, IL, USA). The data were expressed as the means ± SD, and a probability of *p* < 0.05 was considered to be significant.

## 3. Results

### 3.1. Phylogenetic Analysis and Syntenic Location of Elovl8

Phylogenetic analysis was conducted to establish the orthology of the newly identified Elovl8 genes and to resolve previous cases of misidentification due to sequence similarity to Elovl4 genes [29]. To ensure an accurate sorting of our target sequences within the phylogeny, a comprehensive set of Elovl1, Elovl7, Elovl4, and Elovl8 sequences, covering major vertebrate lineages and invertebrate species, were included in the analysis. Elovl8 sequences formed a monophyletic clade outgrouped by Elovl sequences from invertebrate deuterostome sequences. Within the Elovl8 clade, we observed that Teleostei species including *D. rerio*, present two *elovl8* genes grouped into two clades *elovl8a* and *elovl8b*, which most probably resulted from the teleost specific genome duplication (TSGD) event. However, species that diverged prior to the TSGD like the Holostei (e.g., *L. oculatus*) present a single *elovl8* (Figure 1A). Sequence search and collection revealed *elovl8* genes are found in a wide array of vertebrate lineages, while other species still retain remnants of a non-coding *elovl8* gene. More specifically, we identified sequence eroded fragments of an *Elovl8* pseudogene in *H. sapiens* (Figure 1B). We were unable to identify any *Elovl8-like* remnants in two birds (*gallopavo gallus*, *Meleagris gallopavo*), and in three reptiles (*Anolis carolinensis, Alligator mississippiensis*, and *Thamnophis sirtalis*). Synteny analysis showed that the identified remnants of *Elovl8* in *H. sapiens* are located in a genomic location containing the *Kcna* gene cluster. This has also been observed for *B. taurus*, *A. platyrhynchos*, and *C. picta*. On the other hand, the genomic location of the *Elovl8* in amphibians, coelacanth, Teleostei, and Chondrichthyans species is different. In *D. rerio*, we found that *Elovl8* duplicates are located in regions related by TSGD, sharing several conserved genes with other species that did not undergo TSGD, namely *L. oculatus*, *C. milii* and *L. chalumnae.* The different location of the *Elovl8* in mammals, birds, and turtles is possibly due to a translocation of Elovl8 in the amniote ancestor, as the expected location would be in the vicinity of *elovl1* (Appendix A).

### 3.2. Effect of Different Fatty Acids on the Expression of Elovl8a and Elovl8b in ZFL Cells

Next, the mRNA expression levels of both genes were determined in ZFL cells treated with SFAs (C14:0, C16:0, C18:0, C20:0, and C22:0) and PUFAs (C18:2n-6, C18:3n-3, C18:4n-6, C20:3n-3, C20:4n-6, C22:5n-3, and C22:6n-3). For *elovl8a*, incubation of ZFL cells with C18:0 and C20:0 resulted in 1.48-fold change (FC) and 2.33-FC increased expression, respectively (Figure 2A). When PUFAs were supplied to ZFL cells, the expression levels of *elovl8a* showed 3.62-FC, 6.85-FC, 3.73-FC, and 3.13-FC increases when treated with C18:2n-6, C18:3n-3, C20:3n-3, and C22:5n-3, respectively (Figure 2B). Similar to *elovl8a*, *elovl8b* was upregulated in ZFL cells treated with C18:0 (5.71-FC) and C20:0 (1.46-FC), respectively (Figure 2C). However, only C18:2n-6 in PUFA increased the *elovl8b* expression level (1.76 FC) in ZFL cells compared with the control group (Figure 2D).

### 3.3. The Effect of Elovl8a and Elovl8b Knockdown on ZFL Fatty Acid Compositions

In order to gain insight into the roles played by Elovl8a and Elovl8b in FA synthesis, an in vitro study involving knockdown of *elovl8a* and *elovl8b* with specific siRNA (Figure 3A,B) was performed in ZFL cells. After treatment with si:RNA for 48 h, the FA composition of ZFL cells was analyzed. We found that the knockdown of *elovl8a* did not affect the levels of SFAs or MUFAs in ZFL cells (Appendix A). However, si:*elovl8a* treated ZFL cells contained significantly higher levels of C18:2n-6 and C18:3n-3, and lower levels of C20:2n-6 compared to control (Figure 3C). Regarding the si:*elovl8b* treatment, we observed significantly higher levels of C18:0 and C20:0, and lower levels of C22:0 in si:*elovl8b* treated ZFL cells compared to control (Figure 3D). Moreover, significantly lower levels of C18:1 and higher levels of C20:1 compared to controls were detected in si:*elovl8b* treated ZFL cells (Figure 3E). The inhibition of *elovl8b* did not cause any significant changes in PUFA levels (Appendix A).

### 3.4. Expression Pattern of Elovl8a and Elovl8b in Wild Type Zebrafish

qPCR was performed to analyze the expression patterns of zebrafish *elovl8a* and *elovl8b* in different tissues. In adults, *elovl8a* was mainly expressed in the eyes, brain, and liver. The *elovl8b* had a higher abundance transcript level in the ovary, testis, and liver (Appendix A). These patterns were recapitulated in 96 hpf embryos, with a strong *elovl8a* expression in eyes and *elovl8b* in liver and intestine, respectively (Appendix A).

### 3.5. Hepatic Fatty Acid Compositions of Elovl8a^−/−^ and Elovl8b^−/−^ Zebrafish

To investigate the roles of *elovl8a* and *elovl8b* in FA synthesis in zebrafish, both *elovl8a* and *elovl8b* genes were disrupted using the CRISPR/Cas9 technique. Figure 4A shows that the four bases (TTGG) at the target site of *elovl8a* were removed, resulting in a premature stop of translation at codon 54 of the Elovl8a protein. The electropherogram of *elovl8b^−^*^/*−*^ showed a seven-base deletion at the target site, resulting in a premature stop of translation at codon 25 of the Elovl8b protein (Figure 4B). To evaluate the effect of *elovl8a* or *elovl8b* deletion on the embryonic development of zebrafish, we analyzed the survival rates of WT, *elovl8a^−^*^/*−*^ and *elovl8b^−^*^/*−*^ embryos. Our result showed there was no significant difference in the early survival rate of WT, *elovl8a^−^*^/*−*^, and *elovl8b^−^*^/*−*^ embryos (Appendix A).

Next, hepatic FA compositions from WT, *elovl8a^−^*^/*−*^, and *elovl8b^−^*^/*−*^ zebrafish were determined. Similar to the *elovl8a* knockdown experiment, significantly higher levels of 18:2n-6 and 18:3n-3 were observed in *elovl8a^−^*^/*−*^ zebrafish when compared with WT zebrafish. In tandem, a decrease in C20:3n-3 and C22:5n-3 levels were also detected in the *elovl8a^−^*^/*−*^ fish (Figure 4C). Overall, disruption of *elovl8a^−^*^/*−*^ reduced the ratio of the elongation product:substrate (C20:2n-6/C18:2n-6, C20:3n-3/C18:3n-3, and C22:5n-3/C20:5n-3) (Figure 4D). Comparatively, the deletion of *elovl8b* mainly affected the levels of SFAs and MUFAs with increased C18:0 and C20:1 compared to WT zebrafish (Figure 4E). A decreased C20:0/C18:0 ratio was also observed in the *elovl8b^−^*^/*−*^ mutant (Figure 4F). In other words, the results were partially consistent with the *elovl8b* knockdown experiment in vitro and found in *elovl8b^−^*^/*−*^ zebrafish.

### 3.6. Elovl8a and Elovl8b Might Be Involved in Fatty Acid Biosynthesis

In vivo feeding experiment showed that the expression levels of *elovl8a* and *elovl8b* in zebrafish tissues were significantly affected by dietary fatty acid. Compared to the control dietary treatment, the expression levels of *elovl8a* and *elovl8b* significantly increased when fed with high levels of 18:3n-3 and 18:0 or 20:0, respectively (Figure 5A–C). Next, we measured the expression level of PUFA elongation-related genes (*elovl2*, *elovl4s*, *elovl5*) in the liver of *elovl8a^−^*^/*−*^ and the expression level of SFA and MUFA elongation-related genes (*elovl1s*, *elovl3s*, *elovl7s*) in the liver of *elovl8b^−^*^/*−*^. *elovl4b* and *elovl5* exhibited a significantly higher expression level in *elovl8a^−^*^/*−*^ the liver compared to WT (Figure 5D). The expression of *elovl1b* and *elovl3b* exhibited significant upregulation in *elovl8b^−^*^/*−*^ in the liver compared to WT (Figure 5E). In addition, we measured the expression level of *elovl8a* in the liver of *elovl5^−^*^/*−*^ and the expression level of *elovl8b* in the liver of *elovl1a^−^*^/*−*^, *elovl1b^−^*^/*−*^, and *elovl3b^−^*^/*−*^. Results showed higher transcript levels of *elovl8a* in the *elovl5^−^*^/*−*^ strain, while *elovl8b* expression was also increased significantly in the *elovl1a^−^*^/*−*^, *elovl1b^−^*^/*−*^, and *elovl3b^−^*^/*−*^ individuals (Figure 5F–I).

## 4. Discussion

Elovl enzymes catalyze the usually rate-limiting step of the pathway that results in a net two-carbon elongation of pre-existing fatty acyl chains [56]. Elovl1–Elovl7 have been extensively studied in many metazoans including invertebrates and vertebrates. A novel member of the Elovl family, Elovl8, has recently been described in two teleosts [30,31]. However, the exact role of *elovl8* in FA elongation deserves further exploration. In this study, we elucidate the evolutionary history of *elovl8* in metazoans through phylogenetic and comparative synteny analysis and provide critical overarching insights into the in vivo function of *elovl8* by developing knockdown and knockout models in zebrafish.

We identified Elovl8 sequences in chordates including teleosts, amphibians, reptiles, birds, and mammals, a clear indication that *Elovl8* gene is widely distributed in vertebrates and not exclusively unique to teleosts as previously suggested [31]. Tree topology with the invertebrate Elovl8 out-grouping vertebrate deuterostome Elovl8 clade shows that this gene predates the emergence of vertebrates and is not a result of the duplication of an ancestral *Elovl* gene in sarcopterygians [31]. We additionally showed that some tetrapods (*G. gallus*, *M. gallopavo*, *A. carolinensis*, *A. mississippiensis*, and *T. sirtalis*) have lost the *elovl8* gene, while other species like *H. sapiens* still retain a non-coding *elovl8* gene. The direct trigger for loss of the *Elovl8* gene in some of these species requires further investigations.

The elongating function of these two isoforms were deciphered by the measurement of the hepatic FA profile of homozygous teleost *elovl8a*, and *elovl8b* knockout models were constructed by CRISPR/Cas9 technology. The results showed that the deletion of *elovl8a* significantly reduced the levels of C20–C22 PUFAs in zebrafish liver. Specifically, the ratios of C20:2n-6/C18:2n-6 and C20:3n-3/C18:3n-3 decreased significantly, suggesting that *elovl8a* may be involved in the elongation of C18:2n-6 and C18:3n-3. Moreover, *elovl8b* deletion mainly affected the composition of SFAs, which showed the decreased level of the ratio C20:0/C18:0. Thus, *elovl8b* may play an important role in C18:0 elongation. Similar to in vivo experiments, the knockdown *elovl8a* inhibited the elongation of C18:2n-6 and C18:3n-3 to C20:2n-6 and C20:3n-3 in vitro. Moreover, the levels of C18:0 and C20:1 were significantly accumulated in ZFL cells treated with si:*elovl8b*, suggesting that *elovl8b* may be involved in the elongation of C18:0 and C20:1.

In vivo and in vitro experiments confirmed that *elovl8a* activity was specific to C18–C20 PUFAs and *elovl8b* activity was specific to C18:0 and C20:1 MUFAs, suggesting that *elovl8a* had similar functional characteristics to *elovl4* and *elovl5*, and *elovl8b* had similar functional characteristics to *elovl1*, *elovl3*, and *elovl7* [4,8,12,14,26,57]. To further verify the functional overlap of *elovl8a* and *elovl8b* with other elongases, we verified the expression pattern of *elovl8a* in the available Elovl knockout models. The results showed that the expression of *elovl8a* and *elovl8b* were significantly increased in *elovl5^−^*^/*−*^ and *elovl1a^−^*^/*−*^, *elovl1b^−^*^/*−*^, *elovl3b^−^*^/*−*^, respectively, compared with WT.

In conclusion, we established the orthology of the newly identified *elovl* (*elovl8a* and *elovl8b*) and clarified the evolution history of *elovl8* elongases in chordates. Moreover, in this study, a systematic report of elovl8′s elongation functions suggested the substrate preference of Elovl8 overlapped with other Elovls. Our study increased our comprehension of the biochemical pathway for fatty acid biosynthesis (Figure 6).

## Figures and Tables

**Figure 1 genes-12-01287-f001:**
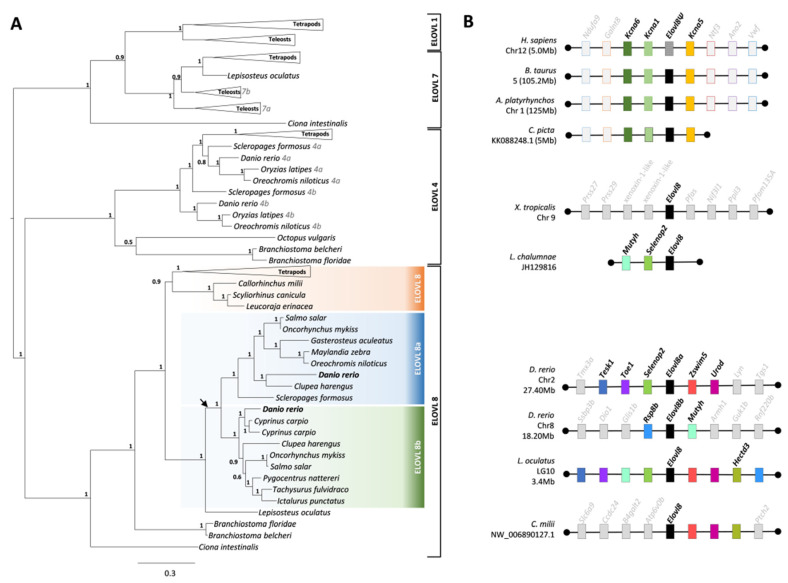
Phylogenetic analysis and syntenic location of *elovl8*. (**A**) Phylogenetic analysis of Elovl1, Elovl7, Elovl4, and Elovl8 sequences; values at node correspond to posterior probabilities provided by aBayes. Tree was rooted at midpoint. (**B**) Syntenic location of the *Elovl8* genes in several species; *Elovl8* gene is represented by black box; dotted black box in human represents a pseudogene; color code of the remaining boxes is conserved corresponding to the same gene identified in several species. Genes identified in a limited number of species with limited or no cross species conservation indicated in grey.

**Figure 2 genes-12-01287-f002:**
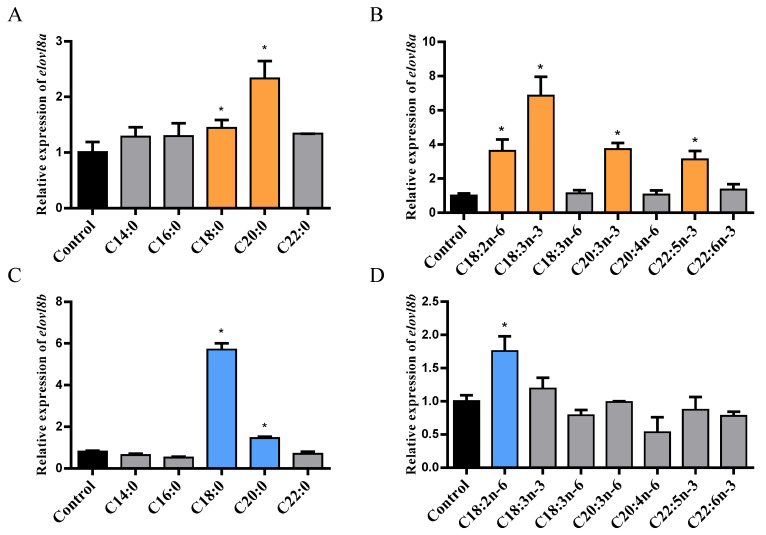
The mRNA expression levels of *elovl8a* and *elovl8b* in different fatty acids treatment zebrafish liver (ZFL) cell. (**A**,**B**) The expression levels of *elovl8a* in ZFL cells supplemented with SFAs (**A**) or PUFAs (**B**). (**C**,**D**) The expression levels of *elovl8b* in SFAs (**C**) or PUFAs (**D**) treatment ZFL cell. The statistical analyses were conducted by *t* test. Data were expressed as mean ± SD (standard deviation) of three biological replicates. The asterisks labeled above the error bars indicated significant differences (* *p* < 0.05). SFAs, saturated fatty acids; MUFAs, monounsaturated fatty acids; PUFAs, polyunsaturated fatty acids; *elovl*, elongation of very long-chain fatty acid protein.

**Figure 3 genes-12-01287-f003:**
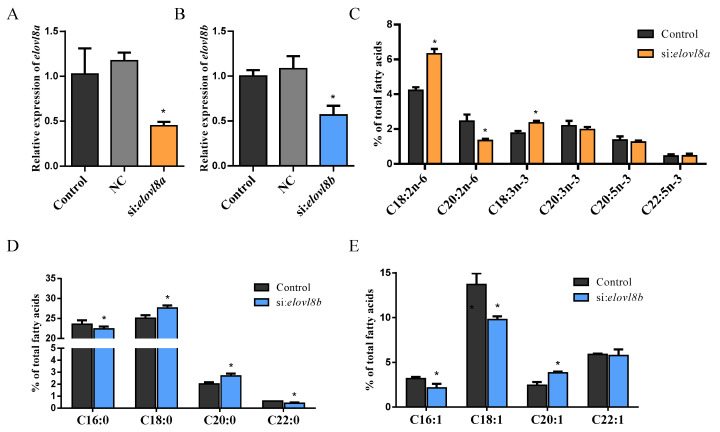
Effects of *elovl8a* and *elovl8b* knockdown on liver fatty acid composition. (**A**) The expression level of *elovl8a* in si:*elovl8a* treated ZFL cells. (**B**) The expression level of *elovl8b* in si:*elovl8b* treated ZFL cells. (**C**) PUFA composition of control and si:*elovl8a* treated ZFL cells. (**D**,**E**) SFA (**D**) and MUFA (**E**) composition of control and si:*elovl8b* treated ZFL cells. The statistical analyses were conducted by *t* test. Data were expressed as mean ± SD (standard deviation) of three biological replicates. The asterisks labeled above the error bars indicate significant differences (* *p* < 0.05). NC, negative control; SFAs, saturated fatty acids; MUFAs, monounsaturated fatty acids; PUFAs, polyunsaturated fatty acids; *elovl*, elongation of very long-chain fatty acid protein.

**Figure 4 genes-12-01287-f004:**
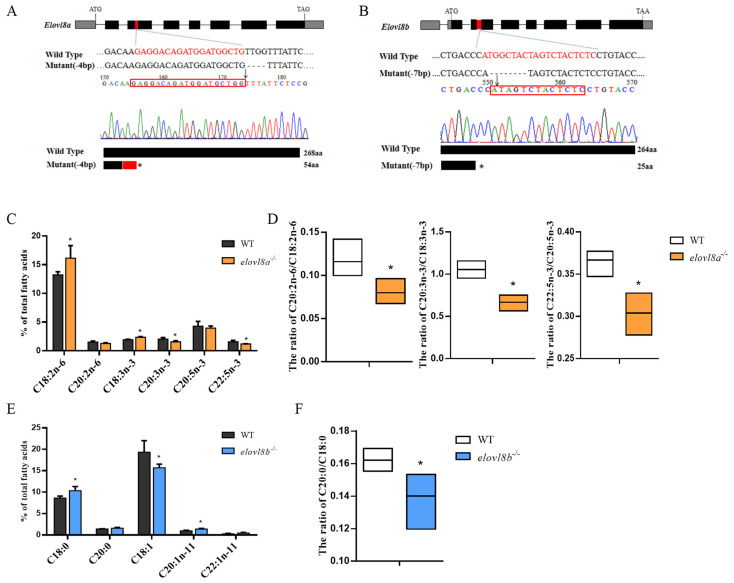
*elovl8a* and *elovl8b* gene deletion in zebrafish and effects of *elovl8a* and *elovl8b* deletion on liver fatty acid compositions. (**A**,**B**) The targeting site for *elovl8a* (**A**) and *elovl8b* (**B**) gene knockout. The gray boxes were the 5′-untranslated region and 3′-untranslated region and black boxes were the exons. The red box indicated the targeting sequences. (**C**) The PUFA composition in liver of wild-type (WT) and *elovl8a^−^*^/*−*^. (**D**) The ratio of C20:2n-6/C18:2n-6, C20:3n-3/C18:3n-3, and C22:5n-3/C20:5n-3 in the liver of WT and *elovl8a^−^*^/*−*^. (**E**) The SFA and MUFA composition in the liver of WT and *elovl8b^−^*^/*−*^. (**F**) The ratio of C20:0/C18:0 in the liver of WT and *elovl8b^−^*^/*−*^. The statistical analyses were conducted by the *t* test. Data were expressed as mean ± SD (standard deviation) of four biological replicates. Asterisks above the error bars indicate significant differences (* *p* < 0.05. aa, amino acid; *elovl*, elongation of very long-chain fatty acid protein; SFAs, saturated fatty acids; MUFAs, monounsaturated fatty acids; PUFAs, polyunsaturated fatty acids.

**Figure 5 genes-12-01287-f005:**
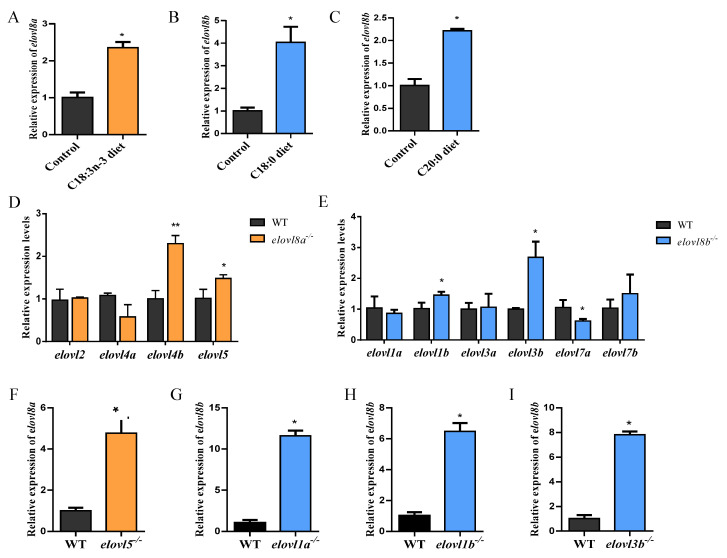
The expression levels of *elovl8a* and *elovl8b* in liver of diet-treatment zebrafish and other elongase knockout zebrafish. (**A**) The expression levels of *elovl8a* in the liver of C18:3n-3 diet-treatment zebrafish. (**B**,**C**) The expression levels of *elovl8b* in the liver of C18:0 and C20:0 diet-treatment zebrafish. (**D**) The expression levels of *elovl2, elovl4s,* and *elovl5* in the liver of *elovl8a* knockout zebrafish (*elovl8a^−^*^/*−*^). (**E**) The expression levels of *elovl1s, elovl3s*, and *elovl7s* in the liver of *elovl8b* knockout zebrafish (*elovl8b^−^*^/*−*^). (**F**) The expression levels of *elovl8a* in the liver of *elovl5* knockout zebrafish (*elovl5^−^*^/*−*^). (**G**–**I**) The expression levels of *elovl8b* in the liver of *elovl1a* knockout zebrafish (*elovl1a^−^*^/*−*^) (**G**), *elovl1b* knockout zebrafish (*elovl1b^−^*^/*−*^) (**H**), and *elovl3b* knockout zebrafish (*elovl3b^−^*^/*−*^) (**I**). The statistical analyses were conducted by the *t* test. Data were expressed as mean ± SD (standard deviation) of three biological replicates. Asterisks above the error bars indicate significant differences (* *p* < 0.05, ** *p* < 0.01). WT, wild type zebrafish; *elovl*, elongation of very long-chain fatty acid protein.

**Figure 6 genes-12-01287-f006:**
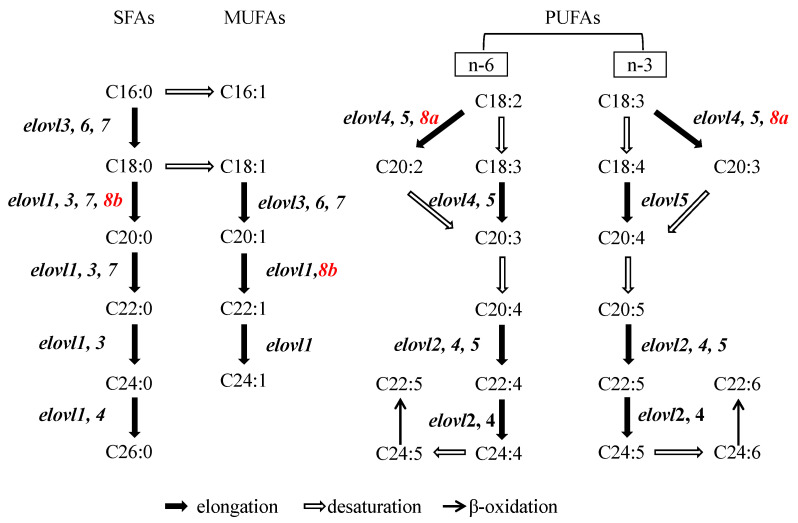
The schematics of biosynthesis pathways of fatty acid synthesis in teleosts. SFAs, saturated fatty acids; MUFAs, monounsaturated fatty acids; PUFAs, polyunsaturated fatty acids; *elovl*, elongation of very long-chain fatty acid protein.

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
