# Peer review of "Evolution and Functional Characteristics of the Novel elovl8 That Play Pivotal Roles in Fatty Acid Biosynthesis"

_genes, 2021, doi:10.3390/genes12081287_

Round 1
Reviewer 1 Report
This manuscript may be of general interest to researchers in the field of FA metabolism.
Minor points:
In general – go through the document and specify acronyms when used first time (SFA, MUFA, etc)
Standardize the acronym of CRISPR/Cas9 (not CRISPER/Cas9 or CRISPR/cas9)
r261. This experiment needs more explanation and/or discussion to clarify why measurements of changes in mRNA expression of in response to addition of different substrates should be an investigation in the substrate-specific elongation activities”. Couldn’t mRNA expression be affected by treatment with many different molecules without meaning that the molecule is a substrate?
r299 Clarify if cells in 3A and 3B are treated with just si:elovl8a in A respective si:elovl8b in B? If separately treated, what was the effect on respective paralog measured in fig 3A and 3B?
Reviewer 2 Report
Sun et al., presented an interesting manuscript entitled “Evolution and functional characteristics of the novel Elovl8 which play pivotal roles in fatty acid biosynthesis” (with ref. number Genes-1290660) regarding the potential evolutionary scenario and their biological role of the two elongation of very long-chain fatty acid (Elovl) 8a and 8b paralogues. The research issue here covered, dissecting the biosynthetic pathway of fatty acids (the very long-chain fatty acids) from two perspectives: its evolution along the different taxa included in the metazoans and their biological function, is interesting. The evolution of elovl genes (in addition to the one of the desaturases) is a key process that might explain the metabolic capacities and nutritional requirements of different metazoans. The characterization of the biosynthetic pathway, where up to 8 different genes act in a sequential manner but with certain redundancy, is also a key knowledge for fundamental and applied (fish farming) biology. Since it is a very complex pathway, characterizing the roles of the 8 different genes requires a very tightly designed experiments as well as the use of elegant approaches such as the ones here applied: in vitro and in vivo nutritional experiments, including gene edition.
In general, although the right tools have been applied, the present reviewer considered that not all the required information and the proper analyses have been conducted to unequivocally address the evolution and origin of elovl8a and b genes as well to decipher its potential role within the biosynthetic pathway of the fatty acids. Furthermore, authors go beyond what their results might show. In particular, a differential gene expression result is not an indicative of the potential substrate preference that the encoded enzyme might have. Different major drawbacks identified in the manuscript lead to ask for a major revision and they can be resumed as:
1-Results here presented do not provide a full image of the evolution of elovl1-8, missing some key components of the gene family. In particular, the elovl2, elovl3, elovl5 and elov6 genes are not included. As it is presented, the phylogenetic tree suggests that elovl8 genes are related to the elovl4 genes, but this might be an artifactual perception of not including the above-mentioned genes elovl2, 3, 5 and 6. In addition to other issues that needs to be addressed (detailed later on).
2-Gene expression analysis by both qPCR and in situ hybridization (ISH) analyses might led to inaccurate conclusions. First, authors need to confirm that the housekeeping gene here used is the most constitutive and stable among the different tissues and nutritional conditions here considered. It seems that authors only considered the use of one housekeeping gene. Regarding ISH images (supplementary data) shows an unspecific staining with the sense probe. Furthermore, results of the qPCR and ISH are not consistent since the higher staining in the brain is not in line with the highest gene expression in the eye (in the case of elovl8a). A detailed description of how ISH was performed should be provided.
3-Gene expression regulation and quantification of FA by HPLC. The use of some concepts and analyses are somehow mislead. For instance, altered gene expression do not necessarily indicate that an enzyme may play an alternative role in FA biosynthesis, it just indicates that its transcription has been activated or inhibited by a stimulus (in this case the addition of a particular FA that might activate the corresponding transcription factor). Furthermore, the altered FAs content when elovl8a and b were knockdown are consistent with the tissues where they are expressed?
4-Some important information is missing. Please, specify the parameters used to search and collect (and keep for further analyses) sequences belonging to the elovl gene families. Also, in each figure legend indicate the number of biological replicates performed. Transfection efficiency in cell lines is not provided. In addition, more detailed information should be provided in the description of gene edition performed. How authors confirm the gene edition was specific and achieved? Do they consider the existence of splicing variants? In all the figures, it is not indicated whether the error bars represent standard deviation or SEM.
5-Authors should compare the gene expression regulation not only with the control treatments but also with the right control, the vehicle treatments, since FAs were previously dissolved in MBCD.
Please, find below the detailed list of issues to be addressed:
L32-33: How a qPCR results indicate that Elovl8a and 8b may play a (specific) role in FA biosynthesis? If one gene is highly expressed upon the addition of a particular molecule (e.g. a specific FA), it is indicative that the transcriptional machinery of this particular gene has been activated. Concluding that this gene might be specifically related to its metabolism is going too far.
L44: in the endoplasmic reticulum
L52: define CoAs
L53: as previously defined, use SFAs and MUFAs abbreviations before C18- to C22
L71: Atlantic salmon
L72: annotated instead of categorized?
L73: termed as elovl8a,b and deposited in genebank
L75: FA has been already defined
L78: in functional assays with yeast
L79: to replicate the FA elongation
L80: obtained from functional assays with yeast…. In addition, something is missing in this sentence. Please, check.
L88: developmental stages
L96: the research work might need an approval from the local ethical committee, right?
L100: please, specify the “efforts” made to minimize the suffering.
L103: coelacanthiforms
L104: sorry, are Ciona intestinalis and the brachiostoma invertebrates? They are deuterostomes, right?
L105: numbers
L113: analyses were performed
L114: the evolutionary model
L119-122: please, include the full-length name when first time a scientific name is cited.
L133: experimental diet…. What about the 48% remaining of the feed formulation?
L145: the zebrafish liver cell line (ZFL) has been properly described and characterized in previous studies? Please, provide a reference.
L156: clear?
L160: please, indicate the purity of the FAs used
L163-164: Please, provide the transfection efficiency achieved in ZFL. Normally, transfection efficiency in cell lines is a major limitation for its use in transfection studies.
L166-167: for RNA isolation instead of qPCR analyses.
L169: isoPius kit
L171: please, provide the range of RNA quality values
L176-177: dissociation curve is missing a step (increasing temperature rate). Please, check it.
L178-179: which housekeeping gene was used for in vitro and in vivo analyses? Also, considered to test at least 3 different reference genes for each approach.
L193: 1.0 µl was manually injected?
L194: FAs instead of fatty acids
L210: Sun et al., 2020 reported the generation of elovl2 and 5 knockouts, but not the ones for elovl1a, 1b and 3b.
L227: genome duplication (TSGD) event
L228: In Figure 1a and supplementary 1B, as previously commented, the set of sequence used for building the phylogenetic tree is incomplete, at least to unequivocally support the relation of elovl8 and elovl4 genes suggested by the authors. In this sense, in addition to include sequence of the other elovl genes not present in the phylogenetic tree, authors should also include the a and b forms of them in the species used. Authors include one paralogue of elovl7a from oryzias but not the other one (the 7b). Why not all the elovl genes from the same species are not included? Why sometimes the paralogue isoform (a or b) is specifically indicated but not in the other cases? Furthermore, why the sequence NP_001029014 from Ciona intestinalis is not included as it is annotated as elovl4 gene? Finally, what is Br in the branch above the one of branchiostoma floridae?
L239: TSGD, namely….
L241:as the expected location
L274: In fig. 2D gene expression upon C18:4n-6, C20:4n-6 and C22:6n-3 seems to be downregulated. Please, confirm it. Please (as for all the figures) indicate the number of biological replicates (n) used. Please, delete NC, negative control since here it was not used.
L288-290: are the results of qPCR and siRNA consistent for C18:2n-6 and C18:3n-3? Authors should also test the gene expression of elovl8a upon exposure to C20:2n-6.
L291: similarly, are results consistent when C22:0 was added?
L297: Fig. 3C and Fig. 4C gene expression regulation under C20:2n-6 is not consistent. Also, regulation under C16:0, C20:0 and C22:0 is not consistent between in vitro and in vivo experiments.
L330-332: the results were partially consistent.
L348-349: which is the control dietary treatment? The one where no FAs were supplemented? Since the FAs were previously dissolved in MBCD, the right control is missing, a diet where the equal amount of MBCD is included in the diet.
L350-359: since the expression of elovl2, 3 and 5 were evaluated in the zebrafish knockouts (due to the potential redundancy), why authors do not include these genes in the phylogenetic analysis in order to support they close related role (with bigger homology than with the other elovl genes)?
L359-362: this are conclusions, not results. In addition, as already mentioned, authors need further experiments to support the notion of these gene being an alternative enzyme.
L378: Figure 6 should be built up based on HPLC FAs content analysis in zebrafish knockouts rather than on the gene expression of the different elovl genes on each knockout.
L386: The gene expression profile reported in the reference 31 (Li et al., Aquaculture 2020, 522, 735127) (elovl8a was highly expressed in heart and spleen, while elovl8b in brain and eye) is not consistent with the results here presented. Although different fish species are compared, their gene expression profile should be highly conserved. Please, discus it. Might be since the present results are based on a single housekeeping gene that might not have the most constitutive expression along the different tissues?
L387: please, be correct on your statements. The actual distribution of elolv8 genes was already provided in the recent article Li et al. (Aquaculture 2020, 522, 735127) as well as its functional role. I agree that Li et al. didn’t present a complete picture, but neither the present article.
L388-391: Please, consider that the present “elucidation” is based on an incomplete (skewed) analysis. See previous comments on this regard.
L397-398: The closer homology and/or functional similarity of elovl8 genes with elovl4 genes might be artifactual since the phylogenetic tree do not included all the elovl genes and the different ones within each species evaluated (although their existence has been reported. Please, see in this regard the comment regarding the need of including all the paralogues (a and b forms) of each elovl gene of the same species).
L399: XX?
L402-404: The potential redundancy of elovl genes might allow the loss of elovl8 genes in some tetrapods, but this is not the trigger. Please, rephrase it.
L405-406: wrong construction. Although…..?
L406-414: Do I miss something? A transformation of elovl8 genes was here done in yeast?
L417-419: gene expression cannot be the unequivocally indication of one enzyme having a substrate preference. Please, do not confound gene expression regulation with substrate specificity and/or preference. Differences in gene expression might be due to the fact that the FAs supplemented might activate a specific transcription factor that controls the elovl8a and 8b genes, but this is not necessarily related with the encoded enzyme being responsible for the metabolization of this FA.
L424-428: inconsistent results from in vitro and in vivo approaches do not support this.
L438-439: as above-mentioned, a complete phylogenetic tree would support this.
L443-444: again, gene expression results do not allow authors to infer about the potential alternative action of elovl8a and 8b as FA biosynthetic enzymes.
L446-447: similar to previous comments, authors should perform a more complete analyses to support this notion.
